# Methods for Engineering Binders to Multi-Pass Membrane Proteins

**DOI:** 10.3390/bioengineering10121351

**Published:** 2023-11-24

**Authors:** Benjamin Thomas, Karuppiah Chockalingam, Zhilei Chen

**Affiliations:** 1Interdisciplinary Graduate Program in Genetics and Genomics, Texas A&M University, College Station, TX 77845, USA; benjamin.thomas.harmony@gmail.com; 2Department of Microbial Pathogenesis and Immunology, Texas A&M University Health Science Center, Bryan, TX 77807, USA; chockali2@tamu.edu

**Keywords:** transmembrane proteins, multi-pass transmembrane proteins, nanodisc, SMALP, panning, directed evolution

## Abstract

Numerous potential drug targets, including G-protein-coupled receptors and ion channel proteins, reside on the cell surface as multi-pass membrane proteins. Unfortunately, despite advances in engineering technologies, engineering biologics against multi-pass membrane proteins remains a formidable task. In this review, we focus on the different methods used to prepare/present multi-pass transmembrane proteins for engineering target-specific biologics such as antibodies, nanobodies and synthetic scaffold proteins. The engineered biologics exhibit high specificity and affinity, and have broad applications as therapeutics, probes for cell staining and chaperones for promoting protein crystallization. We primarily cover publications on this topic from the past 10 years, with a focus on the different formats of multi-pass transmembrane proteins. Finally, the remaining challenges facing this field and new technologies developed to overcome a number of obstacles are discussed.

## 1. Introduction

Multi-pass membrane proteins such as G-protein-coupled receptors (GPCRs) and ion channels represent some of most important molecules in living organisms. They reside at the interface between cells and the outside world and underpin a wide range of fundamental cellular functions, including cellular signaling, nutrient uptake and secretion, intercellular communication, motility, and adhesion. Multi-pass membrane proteins are the most popular targets for small-molecule drugs, underscoring their importance in drug development. Their residence on the cell surface renders membrane protein targets easily accessible by therapeutic proteins, which are often unable to reach intracellular targets due to their membrane-impermeability. However, only three multi-pass membrane proteins—CD20, CCR4 and GPCR 5D—have been successfully targeted by antibodies as of September 2023 [1] (Table 1), highlighting the challenges of engineering biologics against this important class of molecules.

The most common format of biologics for therapeutic applications is that of antibodies. The majority of currently approved antibody therapeutics are templated on the human immunoglobulin G1 (IgG1) scaffold, followed by IgG4 and IgG2 [1] (Figure 1A). Advantages of antibody therapeutics include: (i) high specificity, which reduces the likelihood of off-target effects and minimizes harm to healthy tissues; (ii) long circulation half-life, enabling them to remain active in the body for an extended period and reducing the need for frequent dosing (the circulation half-life of wild-type IgG is 10–21 days, depending on the IgG isotype [2], and can reach 62–73 days after Fc engineering [3]); (iii) immune system engagement through recruitment of immune effector cells to enhance efficacy [4,5]; (iv) high potency, as antibody therapeutics typically exhibit a target binding affinity that is orders of magnitude higher than small-molecule drugs; and, (v) rapid development and production thanks to advances in antibody engineering and production technologies. The first antibody cocktail for treatment of COVID-19 was approved by the FDA in November 2020, which was less than a year after the SARS-CoV-2 outbreak. However, antibody therapeutics suffer from a number of limitations: (i) antibody production requires sophisticated mammalian tissue culture, which is expensive and slow to scale up due to the cost of goods and labor, leading to their high cost (>USD 2000 per dose) and limited global production capacity; (ii) immune effector function mediated by the Fc can sometime negatively impact the therapeutic efficacy (for example, the PD-1 antibody penpulimab was Fc engineered to eliminate Fc-mediated effector functions such as antibody-dependent cell-mediated cytoxicity (ADCC), antibody-dependent cellular phagocytosis (ADCP), and reduced antibody-dependent cytokine release (ADCR) to increase its safety profile [6]); and (iii) limited tissue penetration stemming from their bulky size (~150 kDa).

Several antibody fragments in the forms of fragment antigen-binding region (Fab, one constant and one variable domain of each of the heavy and light chain) and single-chain fragments of variable region (scFv, the two variable domains of the light and heavy chains joined by a flexible linker) have been approved for clinical applications. For example, Abciximab, Ranibizumab, Certolizumab pegol, and Idarucizumab are Fab molecules, while Blinatumomab and Romosozumab are scFvs. In addition to conventional antibodies, which comprise a heavy and a light chain, single-domain antibodies with only a heavy chain, known as nanobodies, have been gaining popularity in recent years. Caplacizumab, a humanized nanobody targeting the von Willebrand factor, was approved in 2019 for treating acquired thrombotic thrombocytopenic purpura [7,8]. In addition to antibodies, many non-antibody scaffolds have been developed for therapeutic applications, such as affibody, DARPin, and fynomer [9]. These non-antibody scaffolds can be engineered to exhibit similar target binding affinity and specificity to antibodies, and are amenable to microbial expression, which can potentially lead to lower drug cost and broader accessibility.

The lack of Fc renders antibody fragments/non-antibody scaffold proteins much smaller than conventional antibodies, with much shorter circulation half-lives, necessitating alternative strategies to achieve therapeutic efficacy. For example, blinatumomab, which is about one third the size of a typical antibody, has a short serum half-life (t_1/2_) of ~2 h [10] and is administered by continuous intravenous infusion for 4 weeks [11]. One strategy to achieve extended serum presence is through subcutaneous administration, which is used with the nanobody caplacizumab for treating acquired thrombotic thrombocytopenia [12]. Another strategy is through piggybacking onto native blood proteins with high circulation half-lives, such as antibody [13,14,15,16] and albumin [17]. Finally, polyethylene glycol (PEG) conjugation, which significantly increases the molecular mass, can extend the half-life of biologics in the circulation by protecting against enzymatic digestion, slowing filtration by the kidneys, and reducing the generation of neutralizing antibodies [18,19].

There exist many technologies to engineer target-specific binders (Figure 1B). One of the most widely used technologies for antibody engineering is the hybridoma, in which antibody-producing B cells from immunized animal are fused with immortal cancerous cell lines (e.g., myeloma cells) to create an immortal hybrid cell line that can produce antibodies indefinitely [20]. Although labor-intensive and time-consuming, hybridoma technology has facilitated the discovery of a large repertoire of rodent-derived antibodies for both diagnostic and therapeutic applications. A major limitation of hybridoma technology is the challenge of producing large quantities of stable human monoclonal antibodies due to the absence of suitable myeloma cell lines [21]. Taking advantage of recombinant DNA technology, several cell-based and cell-free display methods have been developed over the past few decades to provide the linkage of phenotype (protein function) with genotype (mRNA/cDNA) for enriching target-specific binders from libraries of variants. In cell-based methods such as phage/yeast-display [22,23], the binder library is displayed as a fusion to a host cell surface protein. Because the conventional PCR-based method for creating scFv libraries from B cells yields a large percentage of incorrectly paired heavy and light chains, very large libraries typically need to be screened in order to identify functional clones. Phage-display can typically accommodate a larger library size (10^10^–10^11^) than yeast-display (10^7^–10^9^), and has been extensively used for scFv and Fab screening. On the other hand, yeast is a eukaryotic organism, similar to mammalian cells, and can perform post-translational modification and protein folding more accurately than bacteriophages. This makes yeast display especially useful for screening human antibody libraries. The accessible library size for cell-based display technologies is dictated by the transformation efficiency. Unlike cell-based methods, cell-free display methods in which the protein is linked directly to the coding nucleic acid sequence can accommodate much larger library sizes (10^11^–10^13^), and have been successfully used for antibody binder engineering [24]. Of particular note are several recently developed display technologies that covalently link the displayed protein to its coding cDNA, thereby avoiding the need to maintain the integrity of potentially unstable mRNAs [25,26,27]. Compared to binder libraries derived from immunized hosts, which are composed of natural proteins for which a vast landscape of possible sequences have already been explored, synthetic libraries are generated using degenerate oligos (e.g., NNK) with mutational spaces that consist of nonsense mutations (e.g., stop codons) and many poorly plausible mutation combinations. Consequently, unlike immune libraries, which often yield high-affinity binders after screening 10^2^ to 10^3^ hybridoma clones, naïve synthetic libraries often require the screening of much larger libraries, typically on the order of 10^7^–10^11^ clones. Recent development of generative language models trained from millions of natural protein sequences may enable more intelligent design of naïve binder libraries with richer sequence space and better evolutionary fitness [28,29,30].

For engineering antibodies using animal hosts, although rodents have historically been the primary immunization host, and most of the current FDA-approved antibodies are of rodent origin, antibodies and fragments derived from alternative hosts or synthetic libraries have recently been gaining in popularity. Chickens, which diverged from mammals about 300 million years ago, have emerged as a promising host for engineering therapeutic antibodies [31]. Humans and rodents are closely related evolutionarily, and their proteins share high homology. Regions of proteins that are homologous across species tend to be conserved through evolution, and as such are generally functionally important. Thus, cross-species conserved protein epitopes represent potentially promising drug targets. Because self-antigens are not immunogenic for the host, it is very challenging to obtain murine antibodies targeting conserved epitopes from a human protein. On the other hand, chicken proteins share little homology with their human counterparts, enabling chickens to generate more diverse antibody repertoires than rodents, especially those targeting highly conserved epitopes. The recent development of transgenic chickens with humanized immunoglobulin genes enables fully human monoclonal antibodies to be generated [32], and should greatly accelerate chicken antibody development.

Another recently popular host for immunization is camelids, a species that is more distant from humans than rodents and, more importantly, produces single-domain antibodies (nanobodies) consisting of only a variable heavy chain (VHH). Nanobodies are small in size (12~15 kD), exhibit high stability, can have high binding affinities for their target antigens, and are compatible with most protein display technologies. Moreover, the heavy chain-only format dramatically reduces the library size generated from the immune repertoire, allowing high-affinity binders to be rapidly enriched.

In addition to animal immunization, many naïve/synthetic libraries templated on human or synthetic antibody scaffolds have been successfully used with cell-based and cell-free display technologies to yield binders with high therapeutic potential. All display-based engineering technologies require highly purified target protein due to the mechanism of the panning process. During panning, a library of binders is first incubated, with the target protein immobilized on a solid support. The solid support is then thoroughly washed to remove the non-binders. Library members that remain associated with the solid support after washing are eluted and amplified, then undergo subsequent rounds of panning until a desired level of enrichment is achieved. Because binders to everything immobilized on the solid support (e.g., target protein and impurities) are equally amplified and enriched, a highly purified target protein sample is crucial to the success of panning.

Unfortunately, because multi-pass transmembrane proteins are often refractory to purification due to their dependence on a lipid environment for folding and activity, engineering binders to this class of targets is much more challenging than to non-membrane targets, as evidenced by the scarcity of biologics binding to multi-pass membrane proteins. Below, we summarize recent technological advances for preparing membrane proteins as targets for binder engineering (Table 2).

## 2. Formats of Multi-Pass Membrane Proteins for Affinity Selection

### 2.1. Soluble Extracellular Loop Fragment

For multi-pass membrane proteins possessing one or multiple extracellular loops that can fold independently of the transmembrane region, these extracellular loops can be synthesized as soluble proteins and used for affinity selection (Figure 2a). For example, CD20 is a member of the MS4A (membrane-spanning 4-domain family A) protein family, and spans the cell membrane four times with two extracellular domains. CD20 is a conserved marker found on B cells, and is a validated target for diseases stemming from abnormal B cells such as leukemia and autoimmune disease. Using the recombinantly expressed large extracellular loop (amino acid 144–188) as the target, Sham et al. screened a naïve human scFv phage library (Yama I library) and successfully enriched anti-CD20 scFvs after three rounds of panning [33]. The best binder, G7 scFv, specifically bound to CD20 expressed on Raji cells with an estimated affinity of ~64 nM. Similarly, Glumac and co-workers panned a naïve human scFv phage library against the extracellular N terminal (amino acids 1–34) and C-terminal (amino acid 126–275) domains of SARS-CoV-2 ORF3a transmembrane protein, obtaining N3aB02 and 3aCA03 single-chain antibodies that were able to bind the full-length ORF3a in transfected cells [34]. Despite these successes, it is worth noting that many multi-pass transmembrane proteins have small extracellular loops that are dependent on the transmembrane region for proper folding, and as such are not suitable targets for affinity selection using this method.

### 2.2. Detergents

Multi-pass membrane proteins often misfold, denature, or aggregate in aqueous solution in the absence of cell membrane due to the hydrophobicity of their transmembrane regions. For the past 40 years, detergents have been the molecules of choice for solubilizing transmembrane proteins. Detergents are amphiphilic compounds with well-segregated polar and apolar domains, allowing them to simultaneously bind both the hydrophobic and hydrophilic regions of a membrane protein (Figure 2b). However, solubility does not always translate to the native structure, and a detergent useful for protein extraction may not be compatible with the subsequent purification steps and/or biochemical studies. Furthermore, there are currently no universal rules that define detergent compatibility with membrane proteins, and case-by-case optimization is usually needed. In addition, detergent-solubilized membrane proteins sometimes adopt partially non-native conformations, which is troublesome for affinity selection. Consequently, different detergents are sometimes used to solubilize membrane proteins at different steps during affinity selection. For example, Kumar et al. used n-dodecyl b-D-maltoside (b-DDM) to solubilize and purify NorC, a putative 14 transmembrane helix multidrug efflux transporter, from *Staphylococcus aureus* [35]. Purified NorC was then reconstituted into *E. coli* polar lipids to form proteoliposomes and used to immunize a 4-year-old male camel. After six rounds of immunization, a nanobody library was constructed from peripheral blood mononuclear cells, displayed on yeast, and panned against fluorescently labelled NorC. As a result, they obtained the nanobody ICab3 with nanomolar NorC binding affinity [35]. In a separate study, Kaur et al. successfully engineered nanobodies against BamA insertase, a 16-stranded transmembrane β-barrel membrane protein that is an essential member of the Gram-negative bacterial outer membrane [36]. BamA in micelles of N,N-dimethyldodecylamine N-oxide (LDAO) was used to immunize an alpaca (200 μg every 2 weeks for 8 weeks). The resulting nanobody library was displayed on phage particles and panned against BamA solubilized in β-DDM. Twenty-one unique nanobodies were identified from ELISA and three were able to bind BamA in solution.

ELISA-based screening of immune libraries sometimes yields binders with limited diversity. For example, to engineer nanobodies as crystallization chaperones of bacterial ABC transporter TM287/288, Egloff et al. immunized an alpaca with a detergent-purified target protein and carried out two rounds of phage panning followed by ELISA-based screening [37]. Sanger sequencing of 210 ELISA hits yielded only 33 unique nanobody sequences; these were nearly identical, belonging to only five binder families [37]. To improve the chance of identifying more diverse binders, they developed the novel NestLink technology, in which a library of short peptide barcodes (termed flycode) is genetically fused to the library of binders. Next-generation sequencing (NGS) was used to pair each flycode with a binder molecule. After selection, flycodes are proteolytically released from the binder and detected via LC-MS/MS to reveal the identity of the corresponding binder. When using NesLink on the same immune library, 29 binder families of TM287/288 were identified, more than five times the number of families obtained by conventional ELISA-based screening.

To ensure the selection of binders against a native epitope, target proteins in multiple formulations/formats can be used alternately. For example, to engineer antibodies against Na^+^/taurocholate co-transporting polypeptide (NTCP), a key receptor for the hepatitis B virus, Takemori et al. immunized NTCP-knockout mice with NTCP solubilized in detergent (1% DMM), reconstituted in liposomes, and expressed in transfected cells [38]. After 4–6 rounds of immunization, they screened the supernatants of hybridomas using a flow cytometric assay and identified mAb N6HB426, which efficiently inhibited HBV infection by binding to the extracellular domain of NTCP.

In order to bypass the need for immunization, which requires access to animal facilities and is often time-consuming and expensive, Zimmermann et al. created three synthetic single-domain antibody (named sybody) libraries tailored for membrane protein targets [39]. These libraries were designed to mimic the natural shape diversity of camelid nanobodies while exhibiting diverse surfaces with moderate hydrophobicity in the randomized region to match the limited hydrophilic epitopes on membrane proteins. By combining ribosome and phage display, they successfully generated high-affinity sybodies against multiple detergent-solubilized membrane proteins, including TM287/288 and the human SLC transporters GlyT1 and ENT1.

### 2.3. Nanodiscs

Nanodiscs are discoidal lipid bilayers of 8–16 nm in diameter stabilized and rendered soluble in aqueous solution by two encircling belts of amphipathic helical proteins (termed membrane scaffold proteins; see Figure 2c [40]). Membrane proteins of diverse types and topologies have been successfully incorporated into nanodiscs. To prepare nanodiscs, a detergent-solubilized mixture containing the target protein is mixed with a lipid cocktail and the scaffold protein. The size of the nanodiscs is determined by the scaffold protein length, while the yield is impacted by the molar stoichiometric ratio of lipid to scaffold protein. Because the target membrane proteins first need to be solubilized without aggregation, the choice of detergent, speed of detergent removal, and identities of the lipids are critical parameters governing nanodisc formation [41]. Nevertheless, nanodiscs are considered more user-friendly than detergents, as the membrane proteins assembled in nanodiscs are more stable and easier to handle.

To develop nanobodies against human apelin receptor (APJ), a class A G-protein coupled receptor (GPCR) responsible for mediating fluid homeostasis and cardiovascular function, Ma et al. immunized camels with APJ nanodiscs, panned the immune repertoire-displaying phage library against APJ proteoliposomes, and identified nanobody JN241 with high APJ binding affinity (*K*_d_ (dissociation constant) 83 pM) [42]. They subsequently solved the complex structure of APJ-JN241 through crystallography and designed a new nanobody, JN241-9 APJ, capable of antagonizing APJ for the treatment of chronic heart failure. Similarly, Yu et al. isolated nanobodies targeting the influenza matrix-2 (M2) protein, a tetrameric transmembrane proton channel important for virus uncoating in endosomes [43]. M2-nanodiscs were used in both the immunization of *Chiloscyllium plagiosum* (a shark) and as the target for panning of the phage-displayed immune library. The best nanobody, AM2H10, showed specificity for the tetrameric M2 ectodomain but not for the monomeric M2 one, and efficiently blocked ion influx through the M2 channel.

To increase the expression efficiency and nanodisc yield, the N- and C-terminal domains of a target protein may need to be modified. For example, Qiang et al. truncated the N- and C-terminal domains of acid-sensing ion channel 1a (ACIS1a) before assembling them into nanodiscs. Acid-sensing ion channels (ASICs) are important targets for pain and stroke. The resulting ACIS1a nanodiscs were biotinylated and used as the target in the panning of a naïve human scFv phage library. Excess amounts of non-biotinylated empty nanodiscs were included during panning to deselect nanodisc-binding phage particles. Six scFvs emerged from the screen; the best candidate, ASC06-IgG1, recognized hASIC1a in a conformation-dependent manner and dose-dependently inhibited acid-induced opening of the channel [44].

To further optimize the use of nanodiscs in the phage display selection pipeline, Dominik et al. proposed a novel strategy that enhances the accessibility of the target epitope [45]. Instead of biotinylating the target protein–nanodisc complex, which may result in biased presentation of the epitopes, they used biotinylated scaffold protein. This strategy enables both sides of the target protein’s surface to be equally accessible during panning selection. They demonstrated this strategy by panning a library of phage-displayed synthetic antibodies against two model membrane proteins: Mj0480 (a small YidC homolog from *Methanocaldococcus jannaschii*) and CorA (a pentameric magnesium ion channel from *Thermotoga maritima*).

In addition to facilitating library panning against membrane targets, nanodiscs have been used to streamline the high-throughput screening of hybridomas. Conventionally, unsorted hybridomas are plated polyclonally in microtiter wells. Following identification of a positive binding well, the hybridomas are then clonally plated to allow the identification of individual hybridoma clones. This is a time-consuming and labor-intensive process. Taking advantage of the high solubility of membrane proteins assembled in nanodiscs, Gardill et al. developed a fluorescence-activated cell sorting (FACS)-based method to sort monoclonal hybridoma cells early in the screening process. They were able to identify hybridoma clones specific to the multi-pass membrane protein VSD4-NavAb, a chimera of human Nav1.7 and bacterial NavAb [46].

### 2.4. SMALPs

Nanodisc technology requires the membrane proteins to be presolubilized in detergent prior to assembly into the new lipid-containing nanoparticles. This poses a signification limitation, as case-by-case optimization is usually required to formulate the detergent and lipid compositions while preserving the natural structure of the target membrane protein. Styrene maleic acid (SMA) was found to fragment the membrane and create membrane islands encompassing membrane proteins, leading to the formation of SMA lipid particles (SMALPs) [46]. SMALPs contain a central lipid bilayer encased by an outer annulus of the SMA polymer, forming disc-shaped nanoparticles [47] (Figure 2d). The nominal maximal diameter of SMALPs is 15 nm, corresponding to a molecular mass of less than 400 kDa [47,48]. SMALPs represent the only technology currently available that permits the extraction of membrane protein in their native lipid environment.

Examining the literature, we did not find any examples of successful use of SMALPs for high-throughput binder selection. We speculate that this may be due to the technical complexity associated with the chemical synthesis of SMA co-polymer, which might be out of reach for many molecular biology labs, along with the relatively nascent nature of this technology. Nevertheless, SMALPs have been extensively used for characterizing binders to membrane proteins. For example, to identify inhibitors of P-glycoprotein (P-gp), which is highly expressed in cancer cells and responsible for multidrug resistance of anti-cancer drugs, Cao et al. used P-gp SMALP-functionalized surface plasmon resonance (SPR) biosensors to screen fifty natural compounds, identifying five P-gp ligands that increased the cells’ drug susceptibility [49]. To study the signaling of parathyroid hormone receptor 1 (PTH1R), Sarkar, et al. used PTH1R-SMALPs and SPR to investigate its interaction with antibodies targeting the extracellular domain of PTH1R [50,51]. Similarly, Velappan et al. used SMALPs to confirm the ability of engineered antibodies to bind the native cytoplasmic domain of the M2 protein of influenza A [52]. In addition, SMALPs have been successfully used in combination with fluorescence correlation spectroscopy (FCS) to characterize the binding capacity of adenosine A2A receptor, an archetypical GPCR, to its ligand [53].

### 2.5. Virus-Like Particles (VLPs)

The retroviral core protein Gag has the ability to self-assemble and bud from host cells, producing noninfectious membranous virus-like particles (VLPs). Co-expressing a target membrane protein and Gag in mammalian cells yields VLPs harboring the target protein [54]. VLPs are homogenous and physically well-defined, while the target membrane proteins in VLPs are presented in a physiologically relevant topology (Figure 2e). Because retroviruses bud from distinct tetraspanin-rich areas of the cell membrane, membrane proteins naturally located at these sites are most efficiently incorporated into the nascent particle’s envelope [55]. For example, platelet-derived growth factor receptor (PDGFR) is highly enriched on VLPs, and the transmembrane domain (TMD) of PDGFR has been extensively explored for incorporating recombinant proteins onto VLPs [56,57]. Oversized cytoplasmic domains of the membrane proteins, which can cause steric hindrance during virus assembly, can be truncated in order to improve the efficiency of VLP incorporation [58].

To generate antibodies against GLUT4, an insulin-responsive 12-transmembrane transporter, Tucker et al. purified GLUT4-containing murine leukemia virus (MLV)-based VLPs and used them to immunize chickens [59]. The concentration of GLUT4 on VLPs was ~300 pmol/mg total protein, which is ~10–100-fold higher than the concentration on intact cells (0.1–1 pmol/mg). The chicken scFv library was displayed through phage-display and panned against GLU4 VLPs (positive selection) and null VLPs (for deselection), yielding several antibodies able to recognize native GLUT4 in cells with apparent affinities as high as 1 pM. Similarly, VLPs harboring CLDN6, a tumor-associated antigen, were used to identify a panel of anti-CLDN6 mouse and chicken antibodies with high affinity and specificity [60,61]. VLPs are sometimes used in conjunction with other protein formats during affinity selection in order to ensure that the membrane protein is in its native conformation. For example, to engineer nanobodies targeting GPCR glucagon receptor (GCGR), which is a model G-protein-coupled receptor with a small extracellular domain, van der Woning et al. immunized two llamas with DNA encoding GCGR, which was performed four times, followed by a single booster with dromedary Caki cells overexpressing GCGR [62]. After confirming the immune response to GCGR^+^ Caki cells via flow cytometry, a Fab phage library was constructed and panned against both the extracellular domain (ECD) and GCGR-VLPs, yielding ten different families of nanobodies targeting five different epitopes on the ECD of GCGR.

### 2.6. Whole Cells

Whole cells offer the most natural presentation of membrane proteins. However, binder selection to membrane proteins on whole cells is the most challenging due to the high diversity and abundance of host proteins on the cell surface, which often significantly outnumber the target membrane protein (Figure 2f). Whole cells are ideal for enriching binders to a target or targets that are naturally dominant. For example, *Alexandrium minutum*, a neurotoxin-producing planktonic algae, was used directly as the target in the panning of a phage library to select nanobodies for the immune detection of *Alexandrium minutum* [63]. In another study, to generate antibodies able to discriminate between malignant and healthy cells, a phage-displayed scFv library was panned against two closely related gastric cancer cell lines [64]. After four rounds of subtractive panning, fourteen unique clones with preferential cell-binding abilities were identified. Using a similar strategy, Furman, et al. successfully enriched cyclic peptides able to mediate binding and internalization into cells expressing the epidermal growth factor receptor [65].

For membrane proteins with moderate cell-surface abundance, immunization is often used in combination with panning. For example, to engineer antibodies targeting CCR6 chemokine receptor, an important target in inflammatory diseases, Gomez-Melero et al. immunized mice with CCR6-overexpressing murine cells (six times at intervals of 2 weeks, followed by three injections once a month) [66]. Screening the hybridoma supernatant against rat cells overexpressing CCR6 yielded the antibody 1C6, which is able to bind the N-terminal domain of CCR6 and block its signaling (IC_50_ 10.23 nM). In another example, to engineer antibodies targeting CD20, we immunized chickens with chicken HD11 cells overexpressing CD20 (day 1 and day 27) and constructed a chicken Fab phage library [67,68]. After four rounds of sequential positive and negative selection against CD20^+^ and naïve CHO or HEK cells, respectively, we identified four chicken antibodies with high affinity to CD20 and 20–100-fold superior whole-blood B cell depletion ability relative to the clinically used anti-CD20 antibody rituximab.

A concurrent positive–negative selection strategy has been used in a number of studies with great success. For example, to engineer antibodies targeting CCR5, the major co-receptor for HIV entry, Shimoni et al. panned naïve human scFv phage libraries against cells that co-express GFP and CCR5 in the presence of excess control cells lacking CCR5 [69]. CCR5-GFP cells were sorted by FACS and the bound phage particles were eluted. After four rounds of panning, several scFv molecules specific for CCR5 were identified, and the most specific clone was confirmed to bind the second extracellular loop of CCR5. Yang et al. used a similar FACS-based strategy to enrich human scFvs able to bind the human GPCR mu opioid receptor (hMOR), one of four major types of opioid receptors [70]. Target cells overexpressing hMOR and yeast cells displaying a naïve scFv library (2.5 × 10^7^ members) were labeled with different fluorophores and co-incubated in the presence of excess parent cells (not labelled). The yeast-target cell complexes were sorted using FACS. After four rounds of selection, two binders with nanomolar affinity to hMOR were identified.

To increase the success rate of engineering GPCR-targeting antibodies, Twist Biopharma developed a synthetic GPCR-focused antibody phage display library with 10^10^ diversity [71]. This library was designed based on a comprehensive multi-species computational analysis of sequences and structures of all known GPCR ligand interactions. After five rounds of panning against CHO cells overexpressing GPCR GLP-1 (positive selection) and the parent CHO cells (negative selection), the enriched pool was analyzed by NGS. Among the ~100 unique antibodies synthesized and expressed as full-length human IgG2, thirteen clones bound specifically to CHO cells overexpressing GLP-1. Importantly, six and four of these clones contained the GLP-1 and GLP-2 peptide motifs, respectively, validating the advantage of this focused library.

In addition to optimizing the library design smarter analytical methods have been explored for improving affinity-selection of binders to difficult membrane proteins by taking advantage of recent developments in NGS technology and machine learning. For example, Morningstar et al. developed an unsupervised machine learning algorithm to identify the structural trends that contribute to affinity by analyzing the NGS data of phage pools after enrichment [72]. As a proof of concept, they screened for antibodies against Frizzled-7, a key ligand in the Wnt signaling pathway, and identified antibodies with picomolar affinity after only two rounds of selection.

## 3. Conclusions

The preparation and presentation of multi-pass membrane proteins for biochemical assays and as targets for protein binder discovery remains a significant challenge, as evidenced by the scarcity of therapeutic antibodies targeting this therapeutically important class of biological molecules. Diverse approaches to the presentation of these complex proteins for protein binder discovery exist, each with their respective pros and cons. The choice of one method over another for multi-pass membrane protein presentation appears to be largely dictated by accessibility to and proficiency with the needed tools. Although detergents remain the go-to agent for solubilizing membrane proteins, new technologies such as nanodiscs, SMALPs, and VLPs have gained significant popularity in recent years. For membrane proteins with established detergent cocktails that support correct folding, such as those with resolved structures, detergent solubilization alone or in combination with nanodiscs provides the easiest way to prepare the target protein for binder engineering. In addition, nanodiscs are relatively small and homogenous in size, which allows better control over the protein-to-lipid radio while reducing sample heterogeneity. For proteins with unknown detergent compatibility, SMALPs can be used to yield purified proteins within their native lipid environment, obviating the need for detergents. However, the lack of published works that have used SMALPs for high-throughput affinity selection point to a number of potential limitations: (i) poor protein extraction efficiency, which may preclude the incorporation of certain target proteins into SMALPs; (ii) interference of the SMA polymer with the solubilized protein or downstream assays; (iii) variable stability of SMALPs, which may require optimization for individual membrane proteins and experiments; and (iv) limited experience within the research community with regard to this new technology. Only time can tell whether SMALPs will become a go-to method for solubilizing membrane proteins. For studies requiring enriched target proteins rather than highly purified ones, VLP technology offers an effective alternative. Membrane proteins are displayed on VLPs in their native conformation, with a controllable orientation that exposes the epitopes of interests and with multiple copies to enable multivalent interaction conducive to binding avidity during panning. However, certain membrane proteins, especially those with large cytoplasmic domains, may be incompatible with VLP incorporation, requiring complex genetic engineering and manipulation that can be challenging and time-consuming. In addition, the lipid environment of VLPs is distinct from that of the plasma membrane, and may not support the correct folding of certain membrane proteins. Finally, purifying VLPs with correctly displayed membrane proteins involves many steps and can be labor-intensive. The only format that guarantees native conformation of any target membrane proteins is whole-cell display. However, the specific targeting of a desired membrane protein in the context of a whole-cell surface crowded with a myriad of non-target proteins continues to pose a daunting challenge. Focused library design can increase library efficiency, while machine learning may allow more diverse candidates to be analyzed. The future development of an efficient technology to specifically target desired membrane proteins in the context of the whole-cell environment would bypass the need for target protein purification, thereby greatly advancing the field.

**Table 1 bioengineering-10-01351-t001:** FDA approved antibodies targeting multi-pass transmembrane proteins.

Name	Drug	Target	Method	Library	Approval	Note	Refs
Tositumomab-I131-2013	Bexxar	CD20	WC	Hybridoma	2003 ^#^	Labeled with I131	[73,74]
Rituximab	MabThera, Rituxan	CD20	WC	Hybridoma	1997	CD20-binding mediated by 2B8	[75]
Ibritumomab tiuxetan	Zevalin	CD20	WC	Hybridoma	2002	Rituximab with tiuxetan attached	[76]
Ofatumumab (HuMax-CD20)	Arzerra	CD20	WC	Hybridoma	2009	Fully human antibody from transgenic mice	[77]
Obinutuzumab (GA101)	Gazyva, Gazyvaro	CD20	ECL	n/a	2013	Glycoengineered murine antibody B-ly1.	[78]
Ocrelizumab	OCREVUS	CD20	n/a	n/a	2017	Fc engineered, humanized from 2H7	[79]
Ublituximab (LFB-R603)	BRIUMVI	CD20	n/a	Hybridoma	2022	Glycoengineered	[80,81]
Mosunetuzumab (CD20-TDB)	Lunsumio	CD20	n/a	n/a	2022	CD20xCD3 bispecific, CD20-binding mediated by 2H7	[82]
Epcoritamab (GEN3013)	EPKINLY	CD20	WC	Hybridoma	2023	CD20xCD3 bispecific, CD20-binding mediated by 7D8	[83,84]
Glofitamab (RG6026)	Columvi	CD20	n/a	n/a	2023	CD20xCD3 (2:1 format) bispecific	[85]
Mogamulizumab (KW-0761)	Poteligeo	CCR4	ECL	Hybridoma	2018	Glycoengineered	[86,87]
Talquetamab (JNJ-64407564)	TALVEY	GPRC-5D	WC	Hybridoma	2023	GPRC-5DxCD3 bispecific	[88,89]

ECL: extracellular loop; WC: whole cell; n/a: information not available; ^#^: currently withdrawn.

**Table 2 bioengineering-10-01351-t002:** Formats of transmembrane proteins successfully used in binder engineering.

Target	Format	Library	Name	Affinity	Function	Refs
CD20	Extracellular loop	Naïve human scFv phage library	G7	K_D_~64 nM	Marker on B cells	[33]
ORF3a	Extracellular loop	Naïve human scFv phage library	N3aB02 3aCA03	K_D_~nM	Viroporin of SARS-CoV-2	[34]
NorC	Detergent	Yeast library from immunized camel	ICab3	K_d_~nM	Multidrug efflux transporter of *Staphylococcus aureus*	[35,90]
BamA	Detergent	Phage library from immunized alpaca	21 clones	K_d_~nM	Insertase of Gram-negative bacteria	[36]
TM287/288	Detergent	Phage library from immunized alpacas	29 families	K_D_~nM-pM	ATP-binding cassette transporter	[37]
MOMP	Detergent	Synthetic nanobody phage library	5 sybodies	n/a	Major outer membrane protein of *Legionella pneumophila* serogroup 6	[37]
NTCP	Detergent	Mouse hybridoma	N6HB426-20	IC_50_~10 nM	Sodium taurocholate cotransporting polypeptide; HBV/HDV entry receptor	[38]
TM287/288	Detergent	Synthetic nanobody phage library	40 sybodies	IC_50_ 62 nM	ATP-binding cassette transporter	[39]
GlyT1	Detergent	Synthetic nanobody phage library	7 sybodies	K_D_~pM-μM	Glycine transporter 1; roles in diseases of the central and peripheral nervous system	[39]
ENT1	Detergent	Synthetic nanobody phage library	Sb_ENT1#1	K_D_ 40 nM	Equilibrative nucleoside transporter 1; roles in ischemia; biomarker of pancreatic cancer	[39]
APJ	Nanodiscs	Phage library from immunized camel	JN241	K_d_ 83 pM	Human Apelin Receptor; mediates fluid homeostasis and cardiovascular function	[42]
Influenza Matrix-2	Nanodiscs	Phage library from immunized shark	AM2H10	K_D_ 78 nM	Proton channel; required for virus uncoating in endosomes	[43]
ASIC1a	Nanodiscs	Naïve human scFv phage library	ASC06-IgG1	K_d_ 7.9 nM	Key ASIC protein activated in neuronal injury	[44]
Mj0480	Nanodiscs	Synthetic Fab phage library	14 clones	K_D_~nM	YidC homolog from *Methanocaldococcus jannaschii*	[45]
CorA	Nanodiscs	Synthetic Fab phage library	10 clones	K_D_~nM	Magnesium ion channel from *Thermotoga maritima*	[45]
VSD4-NavAb	Nanodiscs	Mouse hybridoma	141B8	n/a	Voltage-sensor domain 4 of human Nav1.7 fused to voltage-gated sodium channel from *Acrobacter butzleri*	[46]
GLUT4	VLP	ScFv phage library from immunized chicken	29 clones	K_d_ pM–nM	Glucose transporter type 4; roles in diabetes and obesity	[59]
CLDN6	VLP	ScFv phage library from immunized chicken	6 mAbs	K_D_ pM–nM	Claudin 6; tumor-associated antigen	[61]
CLDN6	VLP	Mouse hybridoma	Polyclonal sera	n/a	Claudin 6; tumor-associated antigen	[60]
GCGR	VLP	Fab phage library from immunized llamas	10 VH families	K_D_~nM	GPCR glucagon receptor; roles in metabolism and homeostasis	[62]
*Alexandrium minutum*	Whole cell	Pre-immune nanobody phage library	4 clones	n/a	Toxic species of dinoflagellates that can cause paralytic shellfish poisoning	[63]
AGS cells	Whole cell	Semisynthetic human scFv phage library	14 clones	n/a	Cells isolated from patient with gastric cancer	[64]
EGFR	Whole cell	Synthetic peptide phage library	11 peptides	IC_50_~µM	Epidermal growth factor receptor; roles in regulation of cell proliferation, differentiation, and migration	[65]
CCR6	Whole cells	Mouse hybridoma	1C6	IC_50_ 10 nM	C-C chemokine receptor type 6; roles in maintaining leukocyte homeostasis and inflammation	[66]
CD20	Whole cells	Fab phage library from immunized chicken	4 mAbs	EC_50_ 12–30 nM	Cluster of differentiate 20; marker on B cells	[67,68]
CCR5	Whole cell	Naïve scFv phage libraries	5 mAbs	K_D_~4 nM	C-C chemokine receptor type 5; co-receptor of HIV	[69]
hMOR	Whole cell	Naïve scFv yeast libraries	2 clones	K_D_~nM	Human GPCR mu opioid receptor	[70]
GLP-1 R	Whole cells	Synthetic GPCR-focused scFv phage library	TB01-3	IC_50_~5 nM	Glucagon-like peptide-1 receptor; receptor for incretin GLP-1	[71]
Fzd7	Whole cells	Synthetic Fab phage library	3 clones	K_d_~pM	Human Frizzled-7; roles in the Wnt signaling pathway	[72]

## Figures and Tables

**Figure 1 bioengineering-10-01351-f001:**
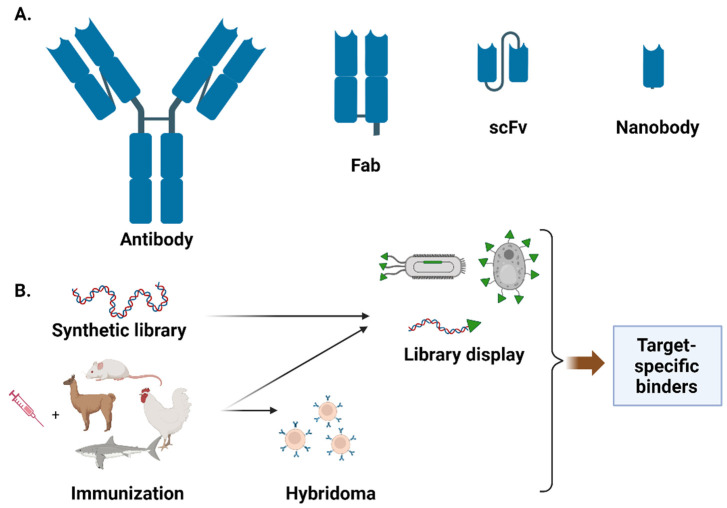
Overview of different binder classes (**A**) and the general strategies for binder engineering (**B**). Figure created with BioRender.com.

**Figure 2 bioengineering-10-01351-f002:**
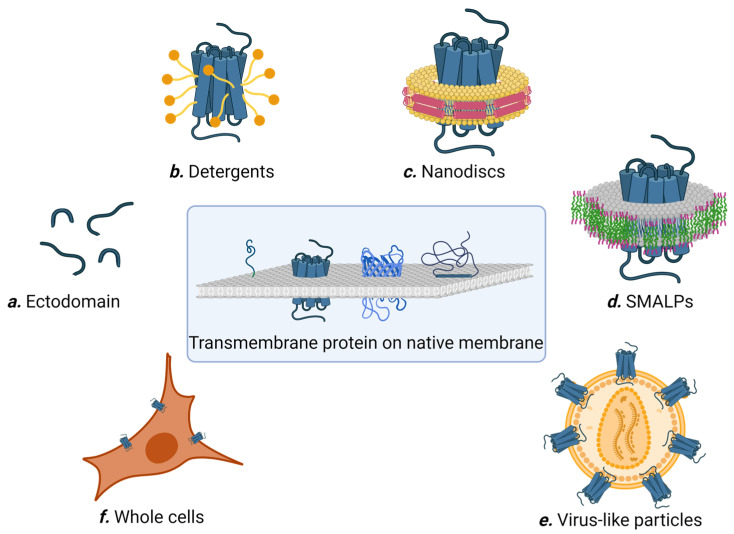
Different methods for preparing membrane proteins for affinity selection. Figure created with BioRender.com.

## Data Availability

Not applicable.

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
