# Peer review of "Methods for Engineering Binders to Multi-Pass Membrane Proteins"

_bioengineering, 2023, doi:10.3390/bioengineering10121351_

Round 1

Reviewer 1 Report

Comments and Suggestions for Authors

This review covers methods for engineering binders to multi-pass membrane proteins. It is well-written, organized, and nicely illustrated. It will be useful for specialists in the field but not likely beyond that. 

Author Response

We thank this reviewer for the kind words. We intend this review to help educate specialists in the field in engineering binders to membrane proteins.

Reviewer 2 Report

Comments and Suggestions for Authors

This review by Thomas and colleagues entitled “Methods for Engineering Binders to Multi-pass Membrane Proteins” is a call to action for drug development against cell surface G-protein-coupled receptors and ion channel multi-pass membrane protein targets. This is an excellent comprehensive review with a focus on the different formats of multi-pass transmembrane proteins.  The authors have covered the literature from research efforts for the development of new biologics to drugs that made it to the market.  The reported challenges and new methodologies used. A table presenting the methods used to generate the FDA-approved drugs will be helpful to the reader.

Author Response

A new table (Table 1) presenting the methods used to generated the FDA-approved antibodies targeting multi-pass transmembrane proteins is included in the revised manuscript.

Reviewer 3 Report

Comments and Suggestions for Authors

The article, titled "Methods for Engineering Binders to Multi-pass Membrane Proteins," addresses the challenges and strategies involved in engineering biologics targeting multi-pass membrane proteins. While it provides a comprehensive overview of various methods and technologies used for this purpose, there are certain areas where the review could be improved.

Firstly, the article effectively highlights the importance and challenges of engineering biologics against multi-pass membrane proteins. It acknowledges the scarcity of successful cases targeting such proteins with antibodies, nanobodies, and synthetic scaffold proteins. However, the review could have benefited from a more explicit and detailed discussion of the limitations and consequences of this scarcity. It's essential to emphasize the significance of this problem in the field of biologics and drug development.

Secondly, the introduction provides a clear background on the importance of multi-pass membrane proteins and the current status of antibody-based therapies. However, it lacks a critical evaluation of the limitations of antibodies as therapeutic agents in this context. For instance, the article could discuss the challenges related to immunogenicity, cross-reactivity, and the potential need for alternative therapeutic approaches beyond antibodies.

The article provides an informative overview of various technologies used for engineering target-specific binders. It effectively describes the advantages and disadvantages of different methods, such as hybridoma technology, cell-based and cell-free display methods, and synthetic libraries. However, it would be beneficial to see a more critical comparison of these methods, including their efficiency, cost-effectiveness, and the specific challenges they address or introduce.

Furthermore, the review could benefit from a more detailed discussion of the recent developments and emerging technologies in the field. It briefly mentions advances in the use of transgenic chickens and synthetic antibody libraries, but these areas deserve more in-depth exploration, particularly regarding their potential to overcome the challenges associated with multi-pass membrane proteins.

While the article highlights the different formats for affinity selection, such as soluble extracellular loop fragments, detergents, nanodiscs, SMALPs, virus-like particles (VLPs), and whole cells, it could provide a more balanced discussion of the practical applicability and limitations of each format. A more detailed analysis of the success rates and challenges associated with these different approaches would enhance the critical review.

In conclusion, the article effectively addresses the challenges and strategies for engineering binders to multi-pass membrane proteins, but it could benefit from a more critical evaluation of the limitations and consequences of the scarcity of successful cases in this area.

Round 2

Reviewer 3 Report

Comments and Suggestions for Authors

The authors have responded to my comments. I believe that the manuscript in its present form is suitable for publication in Bioengineering.